# Differentiable Learning Under Triage

**Nastaran Okati**
MPI for Software Systems
nastaran@mpi-sws.org

**Abir De**
IIT Bombay
abir@cse.iitb.ac.in

**Manuel Gomez-Rodriguez**
MPI for Software Systems
manuelgr@mpi-sws.org

## Abstract

Multiple lines of evidence suggest that predictive models may benefit from algorithmic triage. Under algorithmic triage, a predictive model does not predict all instances but instead defers some of them to human experts. However, the interplay between the prediction accuracy of the model and the human experts under algorithmic triage is not well understood. In this work, we start by formally characterizing under which circumstances a predictive model may benefit from algorithmic triage. In doing so, we also demonstrate that models trained for full automation may be suboptimal under triage. Then, given any model and desired level of triage, we show that the optimal triage policy is a deterministic threshold rule in which triage decisions are derived deterministically by thresholding the difference between the model and human errors on a per-instance level. Building upon these results, we introduce a practical gradient-based algorithm that is guaranteed to find a sequence of predictive models and triage policies of increasing performance. Experiments on a wide variety of supervised learning tasks using synthetic and real data from two important applications—content moderation and scientific discovery—illustrate our theoretical results and show that the models and triage policies provided by our algorithm outperform those provided by several competitive baselines.

## 1  Introduction

In recent years, there has been a raising interest on a new learning paradigm which seeks the development of predictive models that operate under different automation levels—models that take decisions for a given fraction of instances and leave the remaining ones to human experts. This new paradigm has been so far referred to as learning under algorithmic triage [1], learning under human assistance [2, 3], learning to complement humans [4, 5], and learning to defer to an expert [6]. Here, one does not only has to find a predictive model but also a triage policy which determines who predicts each instance.

The motivation that underpins learning under algorithmic triage is the observation that, while there are high-stake tasks where predictive models have matched, or even surpassed, the average performance of human experts [8, 9], they are still less accurate than human experts on some instances, where they make far more errors than average [1]. The main promise is that, by working together, human experts and predictive models are likely to achieve a considerably better performance than each of them would achieve on their own. While the above mentioned work has shown some success at fulfilling this promise, the interplay between the predictive accuracy of a predictive model and its human counterpart under algorithmic triage is not well understood.

One of the main challenges in learning under algorithmic triage is that, for each potential triage policy, there is an optimal predictive model, however, the triage policy is also something one seeks to optimize, as first noted by De et al. [2]. In this context, previous work on learning under algorithmic triage can be naturally differentiated into two lines of work. The first line of work has developed rather general heuristic algorithms that do not enjoy theoretical guarantees [1, 4, 5]. The second has developed algorithms with theoretical guarantees [2, 3, 6], however, they have focused on more

35th Conference on Neural Information Processing Systems (NeurIPS 2021).

restrictive settings. More specifically, De et al. [2, 3] focus on ridge regression and support vector machines and reduce the problem to the maximization of approximately submodular functions and Mozannar and Sontag [6] view the problem from the perspective of cost sensitive learning and introduce a convex and consistent surrogate loss of the objective function they consider.

**Our contributions.** Our starting point is a theoretical investigation of the interplay between the prediction accuracy of supervised learning models and human experts under algorithmic triage. By doing so, we hope to better inform the design of general purpose techniques for training differentiable models under algorithmic triage. Our investigation yields the following insights:

I. To find the optimal triage policy and predictive model, we need to take into account the amount of human expert disagreement, or expert uncertainty, on a per-instance level.

II. We identify under which circumstances a predictive model that is optimal under full automation may be suboptimal under a desired level of triage.

III. Given any predictive model and desired level of triage, the optimal triage policy is a deterministic threshold rule in which triage decisions are derived deterministically by thresholding the difference between the model and human errors on a per-instance level.

Building on the above insights, we introduce a practical gradient-based algorithm that finds a sequence of predictive models and triage policies of increasing performance subject to a constraint on the maximum level of triage. We apply our gradient-based algorithm in a wide variety of supervised learning tasks using both synthetic and real-world data from two important applications—content moderation and scientific discovery. Our experiments illustrate our theoretical results and show that the models and triage policies provided by our algorithm outperform those provided by several competitive baselines[1].

**Further related work.** Our work is also related to the areas of learning to defer and active learning. In learning to defer, the goal is to design machine learning models that are able to defer predictions [10–18]. Most previous work focuses on supervised learning and design classifiers that learn to defer either by considering the defer action as an additional label value or by training an independent classifier to decide about deferred decisions. However, in this line of work, there are no human experts who make predictions whenever the classifiers defer them, in contrast with the literature on learning under algorithmic triage [1–6]—they just pay a constant cost every time they defer predictions. Moreover, the classifiers are trained to predict the labels of all samples in the training set, as in full automation. In active learning, the goal is to find which subset of samples one should label so that a model trained on these samples predicts accurately *any* sample at test time [19–26]. In contrast, in our work, the trained model only needs to predict accurately *a fraction* of samples picked by the triage policy at test time and rely on human experts to predict the remaining samples.

## 2 Supervised Learning under Triage

Let $\mathcal{X} \subseteq \mathbb{R}^m$ be the feature domain, $\mathcal{Y}$ be the label domain, and assume features and labels are sampled from a ground truth distribution $P(\mathbf{x}, y) = P(\mathbf{x})P(y \mid \mathbf{x})$. Moreover, let $\hat{y} = h(\mathbf{x})$ be the label predictions provided by a human expert and assume they are sampled from a distribution $P(h \mid \mathbf{x})$, which models the disagreements amongst experts [27]. Then, in supervised learning under triage, one needs to find:

(i) a triage policy $\pi(\mathbf{x}) : \mathcal{X} \to \{0, 1\}$, which determines who predicts each feature vector—a supervised learning model ($\pi(\mathbf{x}) = 0$) or a human expert ($\pi(\mathbf{x}) = 1$);

(ii) a predictive model $m(\mathbf{x}) : \mathcal{X} \to \mathcal{Y}$, which needs to provide label predictions $\hat{y} = m(\mathbf{x})$ for those feature vectors $\mathbf{x}$ for which $\pi(\mathbf{x}) = 0$.

Here, similarly as in standard supervised learning, we look for the triage policy and the predictive model that result into the most accurate label predictions by minimizing a loss function $\ell(\hat{y}, y)$. More formally, let $\Pi$ be the set of all triage policies, then, given a hypothesis class of predictive models $\mathcal{M}$,

---

[1]Our code and data are available at `https://github.com/Networks-Learning/differentiable-learning-under-triage`

our goal is to solve the following minimization problem[2]:

$$\underset{\pi \in \Pi, m \in \mathcal{M}}{\text{minimize}} \quad L(\pi, m) \qquad \text{subject to} \quad \mathbb{E}_{\mathbf{x}}[\pi(\mathbf{x})] \leq b \tag{1}$$

where $b$ is a given parameter that limits the level of triage, *i.e.*, the percentage of samples human experts need to provide predictions for, and

$$L(\pi, m) = \mathbb{E}_{\mathbf{x}, y, h} \left[ (1 - \pi(\mathbf{x})) \, \ell(m(\mathbf{x}), y) + \pi(\mathbf{x}) \, \ell(h, y) \right]. \tag{2}$$

Here, one might think of replacing $h$ with its point estimate $\mu_h(\mathbf{x}) = \operatorname{argmin}_{\mu_h} \mathbb{E}_{h|\mathbf{x}}[\ell'(h, \mu_h)]$, where $\ell'(\cdot)$ is a general loss function. However, the resulting objective would have a bias term, as formalized by the following proposition[3] for the quadratic loss $\ell'(h, \mu_h) = (h - \mu_h)^2$:

**Proposition 1** *Let $\ell'(\hat{y}, y) = (\hat{y} - y)^2$ and assume there exist $\mathbf{x} \in \mathcal{X}$ for which the distribution of human predictions $P(h \mid \mathbf{x})$ is not a point mass. Then, the function*

$$\overline{L}(\pi, m) = \mathbb{E}_{\mathbf{x}, y} \left[ (1 - \pi(\mathbf{x})) \, \ell'(m(\mathbf{x}), y) + \pi(\mathbf{x}) \, \ell'(\mu_h(\mathbf{x}), y) \right]$$

*is a biased estimate of the true average loss defined in Eq. 2.*

The above result implies that, to find the optimal triage policy and predictive model, we need to take into account the amount of expert disagreement, or expert uncertainty, on each feature vector $\mathbf{x}$ rather than just an average expert prediction.

## 3 On the Interplay Between Prediction Accuracy and Triage

Let $m_0^*$ be the optimal predictive model under full automation, *i.e.*, $m_0^* = \operatorname{argmin}_{m \in \mathcal{M}} L(\pi_0, m)$, where $\pi_0(\mathbf{x}) = 0$ for all $\mathbf{x} \in \mathcal{X}$. Then, the following proposition tells us that, if the predictions made by $m_0^*$ are less accurate than those by human experts on some instances, the model will always benefit from algorithmic triage:

**Proposition 2** *If there is a subset $\mathcal{V} \subset \mathcal{X}$ of positive measure under $P$ such that*

$$\int_{\mathbf{x} \in \mathcal{V}} \mathbb{E}_{y|\mathbf{x}} \left[ \ell(m_0^*(\mathbf{x}), y) \right] \, dP > \int_{\mathbf{x} \in \mathcal{V}} \mathbb{E}_{y, h|\mathbf{x}} \left[ \ell(h, y) \right] \, dP,$$

*then there exists a nontrivial triage policy $\pi \neq \pi_0$ such that $L(\pi, m_0^*) < L(\pi_0, m_0^*)$.*

Moreover, if we rewrite the average loss as

$$L(\pi, m) = \mathbb{E}_{\mathbf{x}} \left[ (1 - \pi(\mathbf{x})) \, \mathbb{E}_{y|\mathbf{x}}[\ell(m(\mathbf{x}), y)] + \pi(\mathbf{x}) \, \mathbb{E}_{y, h|\mathbf{x}} \left[ \ell(h, y) \right] \right],$$

it becomes apparent that, for any model $m \in \mathcal{M}$, the optimal triage policy $\pi_m^* = \operatorname{argmin}_{\pi \in \Pi} L(\pi, m)$ is a deterministic threshold rule in which triage decisions are derived by thresholding the difference between the model and human loss on a per-instance level. Formally, we have the following Theorem:

**Theorem 3** *Let $m \in \mathcal{M}$ be any fixed predictive model. Then, the optimal triage policy that minimize the loss $L(\pi, m)$ subject to a constraint $\mathbb{E}_{\mathbf{x}}[\pi(\mathbf{x})] \leq b$ on the maximum level of triage is given by:*

$$\pi_{m,b}^*(\mathbf{x}) = \begin{cases} 1 & \text{if } \mathbb{E}_{y|\mathbf{x}} \left[ \ell(m(\mathbf{x}), y) - \mathbb{E}_{h|\mathbf{x}} \left[ \ell(h, y) \right] \right] > t_{P,b,m} \\ 0 & \text{otherwise,} \end{cases} \tag{3}$$

*where $t_{P,b,m} = \underset{\tau \geq 0}{\operatorname{argmin}} \, \mathbb{E}_{\mathbf{x}} \left[ \tau \, b + \max \left( \mathbb{E}_{y|\mathbf{x}} \left[ \ell(m(\mathbf{x}), y) - \mathbb{E}_{h|\mathbf{x}}[\ell(h, y)] \right] - \tau, 0 \right) \right].$*

Then, if we plug in Eq. 3 into Eq. 1, we can rewrite our minimization problem as:

$$\underset{m \in \mathcal{M}}{\text{minimize}} \quad L(\pi_{m,b}^*, m) \tag{4}$$

where

$$L(\pi_{m,b}^*, m) = \mathbb{E}_{\mathbf{x}} \left[ \mathbb{E}_{y|\mathbf{x}}[\ell(m(\mathbf{x}), y)] - \text{THRES}_{t_{P,b,m}} \left( \mathbb{E}_{y|\mathbf{x}} \left[ \ell(m(\mathbf{x}), y) - \mathbb{E}_{h|\mathbf{x}}[\ell(h, y)] \right], 0 \right) \right] \tag{5}$$

---

[2]One might think that minimizing over the set of all possible triage policies is not a well-posed problem, *i.e.*, the optimal triage policy is an extremely complex function. However, our theoretical analysis will reveal that the optimal triage policy does have a simple form and its complexity depends on the considered hypothesis class of predictive models (refer to Theorem 3).

[3]All proofs can be found in Appendix A.

with $\text{THRES}_t(x, \text{val}) = \begin{cases} x & \text{if } x > t \\ \text{val} & \text{otherwise.} \end{cases}$

Here, note that, in the unconstrained case, $t_{P,b,m} = 0$ and $\text{THRES}_0(x, 0) = \max(x, 0)$.

Next, building on the above expression, we prove that the optimal predictive model under full automation $m_{\theta_0^*}$ is suboptimal under algorithmic triage as long as the average gradient across the subset of samples which are assigned to the human under the corresponding optimal triage policy $\pi_{m_{\theta_0^*},b}^*$ is not zero. Formally, our main result is the following Proposition:

**Proposition 4** *Let $m_{\theta_0^*}$ be the optimal predictive model under full automation within a hypothesis class of parameterized models $\mathcal{M}(\Theta)$, $\pi_{m_{\theta_0^*},b}^*$ the optimal triage policy for $m_{\theta_0^*}$ defined in Eq. 3 for a given maximum level of triage $b$, and $\mathcal{V} = \{\mathbf{x} \,|\, \pi_{m_{\theta_0^*},b}^*(\mathbf{x}) = 1\}$. If*

$$\int_{\mathbf{x} \in \mathcal{V}} \mathbb{E}_{y|\mathbf{x}} \left[ \nabla_\theta \ell(m_\theta(\mathbf{x}), y)|_{\theta=\theta_0^*} \right] dP \neq \mathbf{0}. \tag{6}$$

*then it holds that $L(\pi_{m_{\theta_0^*},b}^*, m_{\theta_0^*}) > \min_{\theta \in \Theta} L(\pi_{m_\theta,b}^*, m_\theta)$.*

Finally, we can also identify the circumstances under which any predictive model $m_{\theta'}$ within a hypothesis class of parameterized predictive models $\mathcal{M}(\Theta)$ is suboptimal under algorithmic triage:

**Proposition 5** *Let $m_{\theta'}$ be a predictive model within a hypothesis class of parameterized models $\mathcal{M}(\Theta)$, $\pi_{m_{\theta'},b}^*$ the optimal triage policy for $m_{\theta'}$ defined in Eq. 3 for a given maximum level of triage $b$, and $\mathcal{V} = \{\mathbf{x} \,|\, \pi_{m_{\theta'},b}^*(\mathbf{x}) = 0\}$. If*

$$\int_{\mathbf{x} \in \mathcal{V}} \mathbb{E}_{y|\mathbf{x}} \left[ \nabla_\theta \ell(m_\theta(\mathbf{x}), y)|_{\theta=\theta'} \right] dP \neq \mathbf{0}. \tag{7}$$

*then it holds that $L(\pi_{m_{\theta'},b}^*, m_{\theta'}) > \min_{\theta \in \Theta} L(\pi_{m_\theta,b}^*, m_\theta)$.*

The above results will lay the foundations for our practical gradient-based algorithm for differentiable learning under triage in the next section.

## 4 How to Learn Under Triage

In this section, our goal is to find the policy $m_{\theta^*}$ within a hypothesis class of parameterized predictive models $\mathcal{M}(\Theta)$ that minimizes the loss $L(\pi_{m_\theta,b}^*, m_\theta)$ defined in Eq. 4.

To this end, we now introduce a general purpose gradient-based algorithm that first approximates $m_{\theta^*}$ given a desirable maximum level of triage $b$ and then approximates the corresponding optimal triage policy $\pi_{m_{\theta^*},b}^*$[4]. To approximate $m_{\theta^*}$, the main obstacle we face is that the threshold value $t_{P,b,m_\theta}$ in the average loss $L(\pi_{m_\theta,b}^*, m_\theta)$ given by Eq. 5 depends on the predictive model $m_\theta$ which we are trying to learn. To overcome this challenge, we proceed sequentially, starting from the triage policy $\pi_0$, with $\pi_0(\mathbf{x}) = 0$ for all $\mathbf{x} \in \mathcal{X}$, and build a sequence of triage policies and predictive models $\{(\pi_{m_{\theta_t},b}^*, m_{\theta_t})\}_{t=0}^T$. More specifically, in each step $t$, we find the parameters of the predictive model $m_{\theta_t}$ via stochastic gradient descent (SGD) [28], *i.e.*,

$$\begin{aligned} \theta_t^{(j)} &= \theta_t^{(j-1)} - \alpha^{(j-1)} \nabla_\theta \left. L(\pi_{m_{\theta_{t-1}},b}^*, m_\theta) \right|_{\theta=\theta_t^{(j-1)}} \\ &= \theta_t^{(j-1)} - \alpha^{(j-1)} \nabla_\theta \mathbb{E}_{\mathbf{x}} \left[ \pi_{m_{\theta_{t-1}},b}^*(\mathbf{x}) \, \mathbb{E}_{y,h|\mathbf{x}} \left[ \ell(h,y) \right] + (1 - \pi_{m_{\theta_{t-1}},b}^*(\mathbf{x})) \, \mathbb{E}_{y|\mathbf{x}} [\ell(m_\theta(\mathbf{x}), y)] \right]_{\theta=\theta_t^{(j-1)}} \\ &= \theta_t^{(j-1)} - \alpha^{(j-1)} \mathbb{E}_{\mathbf{x}} \left[ (1 - \pi_{m_{\theta_{t-1}},b}^*(\mathbf{x})) \times \mathbb{E}_{y|\mathbf{x}} [\nabla_\theta \, \ell(m_\theta(\mathbf{x}), y)|_{\theta=\theta_t^{(j-1)}}] \right], \end{aligned} \tag{8}$$

where $\alpha^{(j)}$ is the learning rate at iteration $j$. Moreover, the following proposition shows that, under mild conditions, the performance of the triage policies and predictive models improves in each step:

---

[4]At test time, given a predictive model $m_\theta$ and an unseen sample $\mathbf{x}$, we cannot directly evaluate (or, more precisely, estimate using Monte-Carlo) the value of the optimal triage policy $\pi_{m_{\theta^*},b}^*(\mathbf{x})$, given by Eq. 3, since it depends on $\mathbb{E}_{y\,|\,\mathbf{x}}[\cdot]$ and $\mathbb{E}_{h\,|\,\mathbf{x}}[\cdot]$.

**Proposition 6** *Assume that $\nabla_\theta^2 \ell(m_\theta(\mathbf{x}), y) \preccurlyeq \Lambda\mathbb{I}$ for all $\mathbf{x} \in \mathcal{X}$ and $y \in \mathcal{Y}$ and $\alpha^{(j)} < 1/\Lambda$ for all $j > 0$ for some constant $\Lambda > 0$. If, in each step $t$, we find the parameters of the predictive model $m_{\theta_t}$ using Eq. 8, with $\theta_t^{(0)} = \theta_{t-1}$, then, it holds that $L(\pi_{m_{\theta_t}, b}^*, m_{\theta_t}) < L(\pi_{m_{\theta_{t-1}}, b}^*, m_{\theta_{t-1}})$.*

In addition, the following theorem shows that, whenever the loss function $\ell(\cdot)$ is convex with respect to $\theta$, our algorithm enjoys global convergence guarantee:

**Theorem 7** *Let $\ell(\cdot)$ be convex with respect to $\theta$ and the output of the SGD algorithm $\theta_t = \operatorname{argmin}_\theta L(\pi_{m_{\theta_{t-1}}, b}^*, m_\theta)$. Moreover, assume that $\Lambda_{\min}\mathbb{I} \preccurlyeq \nabla_\theta^2 \ell(m_\theta(\mathbf{x}), y) \preccurlyeq \Lambda_{\max}\mathbb{I}$, with $\Lambda_{\min} > 0$, and $\ell(\cdot)$ be H-Lipschitz, i.e., $\ell(m_\theta(\mathbf{x}), y) - \ell(m_{\theta'}(\mathbf{x}), y) \leq H \cdot \|\theta - \theta'\|$. Then, we have that*

$$\lim_{t \to \infty} L(\pi_{m_{\theta_t}, b}^*, m_{\theta_t}) - L(\pi_{m_{\theta^*}, b}^*, m_{\theta^*}) \leq \frac{4H^2 \Lambda_{\max}}{\Lambda_{\min}^2 (1-b)^2}. \tag{9}$$

In practice, given a set of samples $\mathcal{D} = \{(\mathbf{x}_i, y_i, h_i)\}$, we can use the following finite sample Monte-Carlo estimator for the gradient $\nabla_\theta L(\pi_{m_{\theta_{t-1}}, b}^*, m_\theta)$:

$$\nabla_\theta L(\pi_{m_{\theta_{t-1}}, b}^*, m_\theta) = \nabla_\theta \left[ \frac{1}{|\mathcal{D}|} \sum_{i=1}^{|\mathcal{D}|} \ell(m_\theta(\mathbf{x}_i), y_i) - \text{THRES}_{t_{P, b, m_{\theta_{t-1}}}} \left( \ell(m_\theta(\mathbf{x}_i), y_i) - \ell(h_i, y_i), 0 \right) \right]$$

$$= \frac{1}{|\mathcal{D}|} \sum_{i=1}^{\max(\lceil (1-b)|\mathcal{D}| \rceil, p)} \nabla_\theta \ell(m_\theta(\mathbf{x}_{[i]}), y_{[i]})$$

where $\cdot_{[i]}$ denotes the $i$-th sample in increasing value of the difference between the model and the human loss[5] $\ell(m_{\theta_{t-1}}(\mathbf{x}_{[i]}), y_{[i]}) - \ell(h_{[i]}, y_{[i]})$ and $p$ is the number of samples with $\ell(m_{\theta_{t-1}}(\mathbf{x}_{[i]}), y_{[i]}) - \ell(h_{[i]}, y_{[i]}) < 0$.

In the above, we do not have to explicitly compute the threshold $t_{P, b, m_{\theta_{t-1}}}$ nor the triage policy $\pi_{m_{\theta_{t-1}}, b}^*(\mathbf{x}_i)$ for every sample $\mathbf{x}_i$ in the set $\mathcal{D}$, we just need to pick the $\max(\lceil (1-b)|\mathcal{D}| \rceil, p)$ samples with the lowest value of the model loss minus the human loss $\ell(m_{\theta_{t-1}}(\mathbf{x}_{[i]}), y_{[i]}) - \ell(h_{[i]}, y_{[i]})$ using the predictive model $m_{\theta_{t-1}}$ fitted in step $t-1$. To understand why, note that, as long as $t_{P, b, m_{\theta_{t-1}}} > 0$, by definition, $t_{P, b, m_{\theta_{t-1}}}$ needs to satisfy that

$$\frac{d}{d\tau} \left[ \sum_{i \in \mathcal{D}} [\tau b + \max(\ell(m_{\theta_{t-1}}(\mathbf{x}_i), y_i) - \ell(h_i, y_i) - \tau, 0)] \right] \Bigg|_{\tau = t_{P, b, m_{\theta_{t-1}}}} = 0$$

and this can only happen if $\ell(m_{\theta_{t-1}}(\mathbf{x}_i), y_i) - \ell(h_i, y_i) - t_{P, b, m_{\theta_{t-1}}} > 0$ for $\lfloor b|\mathcal{D}| \rfloor$ out of $|\mathcal{D}|$ samples. Here, we are implicitly estimating the optimal triage policies using the observed training labels and human predictions—we are not approximating them using a parameterized model—and, due to Proposition 6, the implementation of the above procedure with Monte-Carlo estimates is guaranteed to converge to a local minimum of the empirical loss. Moreover, note that we can think of the procedure as a particular instance of disciplined parameterized programming [29, 30], where the differentiable convex optimization layer is given by the minimization with respect to the triage policy.

While training each of the predictive models $m_{\theta_t}$, we can implicitly compute the optimal triage policy $\pi_{\theta_t, b}^*$ as described above, however, at test time, we cannot do the same since we would need to observe the label and human prediction of each unseen sample $\mathbf{x}$. To overcome this, after training the last machine model $m_{\theta_T}$, we also fit a model $\hat{\pi}_\gamma(\mathbf{x})$ to approximate $\pi_{m_{\theta_T}, b}^*(\mathbf{x})$ using SGD, *i.e.*,

$$\gamma^{(j)} = \gamma^{(j-1)} - \alpha^{(j-1)} \nabla_\gamma \left[ \sum_{i=1}^{|\mathcal{D}|} \ell'(\hat{\pi}_\gamma(\mathbf{x}_i), \pi_{m_{\theta_T}, b}^*(\mathbf{x}_i)) \right] \Bigg|_{\gamma = \gamma^{(j-1)}}, \tag{10}$$

where $\alpha^{(j)}$ is the learning rate at iteration $j$ and the choice of loss $\ell'(\cdot)$ depends on the model class chosen for $\hat{\pi}_\gamma$. In our experiments, we have found that, using the above procedure, we can

---

[5]Note that, if the set of samples contains several predictions $h_{[i]}$ by different human experts for each sample $\mathbf{x}_{[i]}$, we would use all of them to estimate the (average) human loss.

**Algorithm 1** DIFFERENTIABLE TRIAGE: it returns the weights of a predictive model $m_\theta$ and the weights of a triage policy $\hat{\pi}_\gamma$.

---

**Require:** Set of training samples $\mathcal{D}$, maximum level of triage $b$, number of time steps $T$, number of epochs $N$, mini batches $M$, batch size $B$, learning rate $\alpha$.

1: **function** TRAINMACHINEUNDERTRIAGE($T, \mathcal{D}, M, B, b, \alpha$)
2:      $\theta^{(0)} \leftarrow$ INITIALIZETHETA()
3:      **for** $t = 1, \ldots, T$ **do**
4:          $\theta_t \leftarrow$ TRAINMODEL($\theta_{t-1}, \mathcal{D}, M, B, b, \alpha$)
5:      $\gamma \leftarrow$ APPROXIMATETRIAGEPOLICY($\theta_T, \mathcal{D}, N, M, B, b, \alpha$)
6:      **return** $\theta_T, \gamma$

7: **function** TRIAGE($\mathcal{D}, b, \theta$)
8:      $p \leftarrow$ number of instances in $\mathcal{D}$ with $\ell(m_\theta(\mathbf{x}), y) - \ell(h, y) < 0$
9:      $\mathcal{D}' \leftarrow \emptyset$
10:      **for** $i = 1, \ldots, \max((1 - b)|\mathcal{D}|, p)$ **do**
11:          $\mathcal{D}' \leftarrow \mathcal{D}' \cup \{i\text{-th sample from } \mathcal{D} \text{ in increasing value of } \ell(m_\theta(\mathbf{x}), y) - \ell(h, y)\}$
12:      **return** $\mathcal{D}'$

13: **function** TRAINMODEL($\theta', \mathcal{D}, M, B, b, \alpha$)
14:      $\theta^{(0)} \leftarrow \theta'$
15:      **for** $i = 0, \ldots, M - 1$ **do**
16:          $\mathcal{D}^{(i)} \leftarrow$ the i'th mini batch of $\mathcal{D}$
17:          $\mathcal{D}^{(i)} \leftarrow$ TRIAGE($\mathcal{D}^{(i)}, b, \theta^{(i)}$)
18:          $\nabla \leftarrow 0$
19:          **for** $(\mathbf{x}, y, h) \in \mathcal{D}^{(i)}$ **do**
20:              $\nabla \leftarrow \nabla + \nabla_\theta\, \ell(m_\theta(\mathbf{x}), y)|_{\theta = \theta^{(i)}}$
21:          $\theta^{(i+1)} \leftarrow \theta^{(i)} - \alpha \frac{\nabla}{B}$
22:      **return** $\theta^{(M)}$

23: **function** APPROXIMATETRIAGEPOLICY($\theta, \mathcal{D}, N, M, B, b, \alpha$)
24:      $\gamma^{(M)} \leftarrow$ INITIALIZEGAMMA ()
25:      **for** $j = 1, \ldots, N$ **do**
26:          $\gamma^{(0)} \leftarrow \gamma^{(M)}$
27:          **for** $i = 0, \ldots, M - 1$ **do**
28:              $\mathcal{D}^{(i)} \leftarrow$ the i'th mini batch of $\mathcal{D}$
29:              $\nabla \leftarrow 0$
30:              **for** $(\mathbf{x}, y, h) \in \mathcal{D}^{(i)}$ **do**
31:                  $\nabla \leftarrow \nabla + \nabla_\gamma \ell'(\hat{\pi}_\gamma(\mathbf{x}), \pi^*_{m_\theta, b}(\mathbf{x}))\big|_{\gamma = \gamma^{(i)}}$
32:              $\gamma^{(i+1)} \leftarrow \gamma^{(i)} - \alpha \frac{\nabla}{B}$
33:      **return** $\gamma^{(M)}$

---

approximate well the optimal triage policy $\pi_{m_{\theta_T}, b}$. However, we would like to note that this problem can also be viewed as finding an estimator for the $\alpha$-superlevel set $C^\alpha(f) = \{\mathbf{x} \in \mathcal{X} : f(\mathbf{x}) \geq \alpha\}$ of the function $f(\mathbf{x}) = \mathbb{E}_{y \mid \mathbf{x}}[\ell(m_{\theta_T}(\mathbf{x}), y) - \mathbb{E}_{h \mid \mathbf{x}}[\ell(h(\mathbf{x}), y)]]$ with $\alpha = t_{P, b, m_{\theta_T}}$ from a set of noisy observations. Under this view, it might be possible to derive estimators with error performance bounds building upon recent work on level set estimation [31, 32]. which is left as future work. Refer to Algorithm 1 for a pseudocode implementation of the overall gradient-based algorithm, which returns $\theta_T$ and $\gamma$. Appendix B provides a detailed scalability analysis, which suggests that our algorithm does not significantly increase the computational complexity of vanilla SGD.

## 5 Experiments on Synthetic Data

In this section, our goal is to shed light on the theoretical results from Section 3. To this end, we use our gradient-based algorithm in a simple regression task in which the optimal predictive model under full automation is suboptimal under algorithmic triage.[6]

---

[6]All algorithms were implemented in Python 3.7 and ran on a V100 Nvidia Tesla GPU with 32GB of memory.

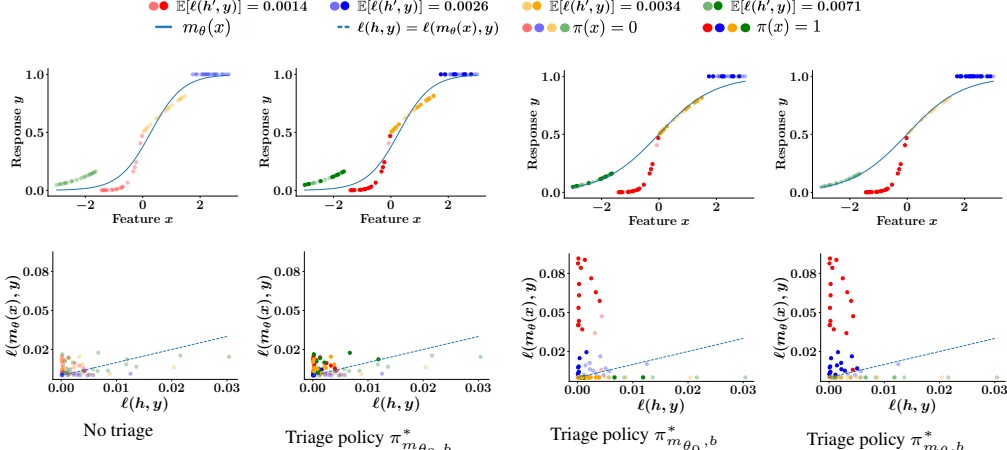

(a) Predictive model $m_{\theta_0}$ trained under full automation    (b) Predictive model $m_\theta$ trained under triage

Figure 1: Interplay between the per-instance accuracy of predictive models and experts under different triage policies. In both panels, the first row shows the training samples $(x, y)$ along with the predictions made by the models $m(x)$ and the triage policy values $\pi(x)$ and the second row shows the predictive model loss $\ell(m(x), y)$ against the human expert loss $\ell(m(x), y)$ on a per-instance level. Columns correspond to the settings (1–4) from left to right. The triage policy $\pi^*_{m_{\theta_0}, b}$ is optimal for the predictive model $m_{\theta_0}$ and the triage policy $\pi^*_{m_\theta, b}$ is optimal for the predictive model $m_\theta$. Each point corresponds to one instance and, for each instance, the color indicates the amount of noise in the predictions by experts, as given by Eq. 11, and the tone indicates the triage policy value. In all panels, we used $\ell(\hat{y}, y) = (\hat{y} - y)^2$ and the class of predictive models parameterized by sigmoid functions, *i.e.*, $m_\theta(x) = S_\theta(x)$.

**Experimental setup.** We generate $|\mathcal{D}| = 72$ samples, where we first draw the features $x \in \mathbb{R}$ uniformly at random, *i.e.*, $x \sim \text{U}[-3, 3]$, and then obtain the response variables $y$ using two different sigmoid functions $S_\theta(x) = \frac{1}{1 + \exp(-\theta x)}$. More specifically, we set $y = S_1(x)$ if $x \in [-3, -1.5) \cup [0, 1.5)$ and $y = S_5(x)$ if $x \in [-1.5, 0) \cup [1.5, 3]$. Moreover, we assume human experts provide noisy predictions of the response variables, *i.e.*, $h(x) = y + \epsilon(x)$, where $\epsilon(x) \sim \mathcal{N}(0, \sigma^2_\epsilon(x))$ with

$$\sigma^2_\epsilon(\mathbf{x}) = \begin{cases} 8 \times 10^{-3} & \text{if} \quad x \in [-3, -1.5) \\ 1 \times 10^{-3} & \text{if} \quad x \in [-1.5, 0) \\ 4 \times 10^{-3} & \text{if} \quad x \in [0, 1.5) \\ 2 \times 10^{-3} & \text{if} \quad x \in [1.5, 3] \end{cases} \tag{11}$$

In the above, we are using heteroscedastic noise motivated by multiple lines of evidence that suggest that human experts performance on a per instance level spans a wide range [1, 2, 27]. Then, we consider the hypothesis class of predictive models $\mathcal{M}(\Theta)$ parameterized by sigmoid functions, *i.e.*, $m_\theta(x) = S_\theta(x)$, and utilize the sum of squared errors on the predictions as loss function, *i.e.*, $\ell(\hat{y}, y) = (\hat{y} - y)^2$, to train the models and triage policies in the following four settings:

1. Predictive model trained under full automation $m_{\theta_0}$ without algorithmic triage, *i.e.*, $\pi_{m_{\theta_0}, b}(\mathbf{x}) = \pi_0(\mathbf{x}) = 0$ for all $\mathbf{x} \in \mathcal{X}$.
2. Predictive model trained under full automation $m_{\theta_0}$ with optimal algorithmic triage $\pi^*_{m_{\theta_0}, b}$.
3. Predictive model trained under algorithmic triage $m_\theta$, with $b = 1$, with suboptimal algorithmic triage $\pi^*_{m_{\theta_0}, b}$. Here, we use the triage policy that is optimal for the predictive model trained under full automation.
4. Predictive model trained under algorithmic triage $m_\theta$, with $b = 1$, with optimal algorithmic triage $\pi^*_{m_\theta, b}$.

In all the cases, we train the predictive models $m_{\theta_0}$ and $m_\theta$ using our method with $b = 0$ and $b = 1$, respectively. Finally, we investigate the interplay between the accuracy of the above predictive models and the human experts and the structure of the triage policies at a per-instance level.

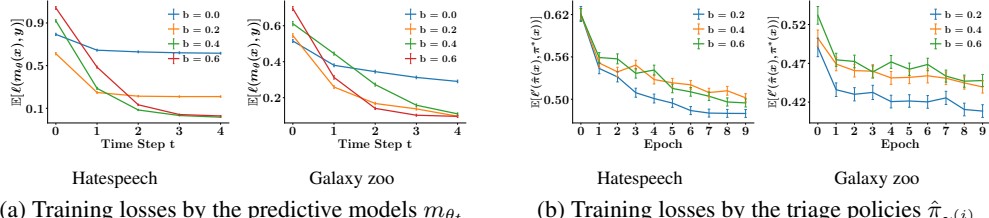

(a) Training losses by the predictive models $m_{\theta_t}$      (b) Training losses by the triage policies $\hat{\pi}_{\gamma^{(i)}}$

Figure 2: Average training losses achieved by the predictive models $m_{\theta_t}$ and the triage policies $\hat{\pi}_{\gamma^{(i)}}$ on the Hatespeech and Galaxy zoo datasets during training. In Panel (a), each predictive model $m_{\theta_t}$ is the output of TRAINMODEL$(\cdot)$ at step $t$ and in Panel (b), each triage policy $\hat{\pi}_{\gamma^{(i)}}$ is the output of TRAINTRIAGE$(\cdot)$ at epoch $i$. Both functions are defined in Algorithm 1. Error bars correspond to plus and minus one standard error.

**Results.** Figure 1 shows the training samples $(x, y)$ along with the predictions made by the predictive models $m_{\theta_0}$ and $m_\theta$ and the values of the triage policies $\pi_0(x)$, $\pi^*_{m_{\theta_0}, b}$ and $\pi^*_{m_\theta, b}$, as well as the losses achieved by the models and triage policies (1-4) on a per-instance level. The results provide several interesting insights.

Since the predictive model trained under full automation $m_{\theta_0}$ seeks to generalized well across the entire feature space, the loss it achieves on a per-instance level is never too high, but neither too low, as shown in the left column of Panel (a). As a consequence, this model under no triage achieves the highest average loss among all alternatives, $L(\pi_0, m_{\theta_0}) = 0.0053$ (setting 1). This may not come as a surprise since the mapping between feature and response variables does not lie within the hypothesis class of predictive models used during training. However, since the predictions by human experts are more accurate than those provided by the above model in some regions of the feature space, we can deploy the model with the optimal triage policy $\pi^*_{\theta_0, b}$ given by Theorem 3 and lower the average loss to $L(\pi^*_{\theta_0, b}, m_{\theta_0}) = 0.0020$ (setting 2), as shown in the right column of Panel (a) and suggested by Proposition 2.

In contrast with the predictive model trained under full automation $m_{\theta_0}$, the predictive model trained under triage $m_\theta$ learns to predict very accurately the instances that lie in the regions of the feature space colored in green and yellow but it gives up on the regions colored in red and blue, where its predictions incur a very high loss, as shown in Panel (b). However, these latter instances where the loss would have been the highest if the predictive model had to predict their response variables $y$ are those that the optimal triage policy $\pi^*_{m_\theta, b}$ hand in to human experts to make predictions. As a result, this predictive model under the optimal triage policy does achieve the lowest average loss among all alternatives, $L(\pi^*_{\theta, b}, m_\theta) = 0.0009$ (setting 4), as suggested by Propositions 4 and 5.

Finally, our results also show that deploying the predictive model $m_\theta$ under a suboptimal triage policy may actually lead to a higher loss $L(\pi^*_{\theta_0, b}, m_\theta) = 0.0031$ (setting 3) than the loss achieved by the predictive model trained under full automation $m_{\theta_0}$ with its optimal triage policy $\pi^*_{\theta_0, b}$. This happens because the predictive model $m_\theta$ is trained to work well *only* on the instances $x$ such that $\pi^*_{m_\theta, b}(x) = 0$ and not necessarily on those with $\pi^*_{m_0, b}(x) = 0$.

## 6 Experiments on Real Data

In this section, we use our gradient-based algorithm in two binary and multi-class classification tasks in content moderation and scientific discovery,. We first investigate the interplay between the accuracy of the predictive models and human experts and the structure of the optimal triage policies at different steps of the training process. Then, we compare the performance of our algorithm with several competitive baselines.

**Experimental setup.** We use two publicly available datasets [33, 34], one from an application in content moderation and the other for scientific discovery[7]:

---

[7]We chose these two particular applications because the corresponding datasets are among the only publicly available datasets that we found containing multiple human predictions per instance, necessary to estimate the human loss at an instance level, and a relatively large number of instances. The Hatespeech dataset is publicly available under MIT license and the Galaxy zoo dataset is publicly available under

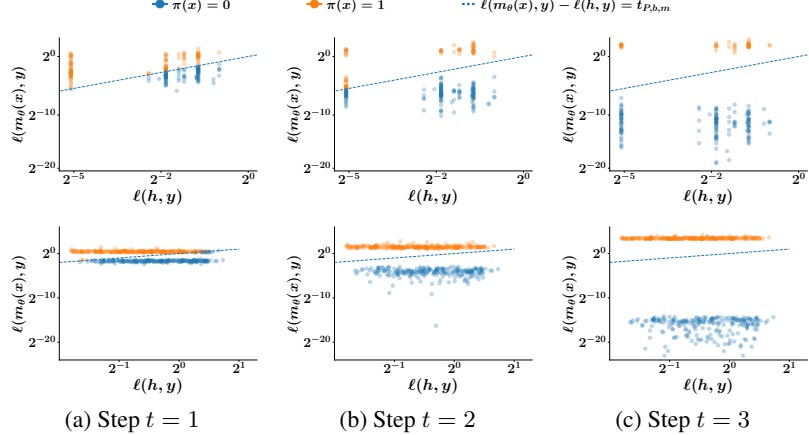

(a) Step $t = 1$        (b) Step $t = 2$        (c) Step $t = 3$

Figure 3: Predictive model and expert losses at a per-instance level on a randomly selected subset of 500 samples of the Hatespeech (top row) and Galaxy zoo (bottom row) datasets throughout the execution of our method during training. The maximum level of triage is set to $b = 0.4$ for Hatespeech and $b = 1.0$ for Galaxy zoo dataset. Each point corresponds to an individual instance and, for each instance, the color pattern indicates the triage policy value.

— *Hatespeech:* It consists of $|\mathcal{D}| = 24{,}783$ tweets containing lexicons used in hate speech. Each tweet is labeled by three to five human experts from Crowdflower as "hate-speech", "offensive", or "neither".

— *Galaxy zoo:* It consists of $|\mathcal{D}| = 10{,}000$ galaxy images[8]. Each image is labeled by $30+$ human experts as "early type" or "spiral".

For each tweet in the Hatespeech dataset, we first generate a 100 dimensional feature vector using fasttext [35] as $\mathbf{x}$, similarly as in De et al. [2]. For each image in the Galaxy zoo dataset, we use its corresponding pixel map[9] as $\mathbf{x}$. Given an instance with feature value $\mathbf{x}$, we estimate $P(h \,|\, \mathbf{x}) = \frac{n_{\mathbf{x}}(h)}{\sum_{h' \in \mathcal{Y}} n_{\mathbf{x}}(h')}$, where $n_{\mathbf{x}}(h)$ denotes the number of human experts who predicted label $h$ and we set its true label to $y = \operatorname{argmax}_{h \in \mathcal{Y}} P(h \,|\, \mathbf{x})$. Moreover, at test time, for each instance that the triage policy assigns to humans, we sample $h \sim P(h \,|\, \mathbf{x})$.

In all our experiments, we consider the hypothesis class of probabilistic predictive models $\mathcal{M}(\Theta)$ parameterized by softmax distributions, *i.e.*,

$$m_\theta(\mathbf{x}) \sim p_{\theta;\mathbf{x}} = \text{Multinomial}\left(\left[\exp\left(\phi_{y,\theta}(\mathbf{x})\right)\right]_{y \in \mathcal{Y}}\right),$$

where, for the Hatespeech dataset, $\phi_{\bullet,\theta}$ is the convolutional neural network (CNN) by Kim [36] and, for the Galaxy zoo dataset, it is the deep residual network by He et al. [37]. During training, we use a cross entropy loss on the observed labels, *i.e.*, $\ell(\hat{y}, y) = -\log P(\hat{y} = y \,|\, \mathbf{x})$. Here, if an instance is assigned to the predictive model, we have that

$$P(\hat{y} = y \,|\, \mathbf{x}) = \frac{\exp\left(\phi_{y,\theta}(\mathbf{x})\right)}{\sum_{y'} \exp\left(\phi_{y',\theta}(\mathbf{x})\right)},$$

and, if an instance is assigned to a human expert, we have that $P(\hat{y} = y \,|\, \mathbf{x}) = P(h = y \,|\, \mathbf{x})$. For the function $\hat{\pi}_\gamma(\mathbf{x})$, we use the class of logistic functions, *i.e.*, $\hat{\pi}_\gamma(\mathbf{x}) = \frac{1}{1 + \exp(-\phi_\gamma(\mathbf{x}))}$, where $\phi_\gamma$ is the same CNN and deep residual network as in the predictive model, respectively. Here, we also use the cross entropy loss, *i.e.*, $\ell'(\hat{\pi}_\gamma(\mathbf{x}), \pi^*_{m_{\theta_T}, b}(\mathbf{x})) = -\pi^*_{m_{\theta_T}, b}(\mathbf{x}) \log \hat{\pi}_\gamma(\mathbf{x}) - (1 - \pi^*_{m_{\theta_T}, b}(\mathbf{x})) \log(1 - \hat{\pi}_\gamma(\mathbf{x}))$. In each experiment, we used $60\%$ samples for training, $20\%$ for validation and $20\%$ for testing. Refer to Appendix C for additional details on the experimental setup.

**Results.** First, we look at the average loss achieved by the predictive models $m_{\theta_t}$ and triage policies $\hat{\pi}_{\gamma^{(i)}}$ throughout the execution of our method during training. Figure 2 summarizes the results, which

---

Creative Commons Attribution-Noncommercial-No Derivative Works 2.0 license. Since our algorithm is general, it may be useful in other applications in which predictive models trained under full automation performs worse than humans in some instances (see Proposition 2).

[8]The original Galaxy zoo dataset consists of $61{,}577$ images, however, we report results on a randomly chosen subset of $10{,}000$ images due to scalability reasons. We found similar results in other random subsets.

[9]The pixel maps for each image are available at `https://www.kaggle.com/c/galaxy-zoo-the-galaxy-challenge`

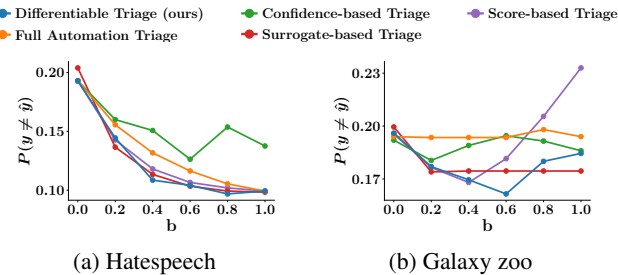

Figure 4: Misclassification test error $P(\hat{y} \neq y)$ against the triage level $b$ on the Hatespeech and Galaxy zoo datasets for our algorithm, confidence-based triage [5], score-based triage [1], surrogate-based triage [6] and full automation triage. Appendix C contains more details on the baselines.

reveal several insights. For small values of the triage level, the models $m_{\theta_t}$ aim to generalize well across a large portion of the feature space. As a result, they incur a large training loss, as shown in Panel (a). In contrast, for $b \geq 0.4$, the models $m_{\theta_t}$ are trained to generalize across a smaller region of the feature space, which leads to a considerably smaller training loss. However, for such a high triage level, the overall performance of our method is also contingent on how well $\hat{\pi}_\gamma$ approximates the optimal triage policy. Fortunately, Panel (b) shows that, as epochs increase, the average training loss of $\hat{\pi}_{\gamma^{(j)}}$ decreases. Appendix D validates further the trained approximate triage policies $\hat{\pi}_\gamma$.

Next, we compare the predictive model and the human expert losses per training instance throughout the execution of the our method during training. Figure 3 summarizes the results. At each step $t$, we find that the optimal triage policies $\pi^*_{m_{\theta_t}, b}$ hands in to human experts those instances (in orange) where the loss would have been the highest if the predictive model had to predict their response variables $y$. Moreover, at the beginning of the training process (*i.e.*, low step values $t$), since the predictive model $m_{\theta_t}$ seeks to generalize across a large portion of the feature space, the model loss remains similar across the feature space. However, later into the training process (*i.e.*, high step values $t$), the predictive models $m_{\theta_t}$ focuses on predicting more accurately the samples that the triage policy hands in to the model, achieving a lower loss on those samples, and gives up on the remaining samples, where it achieves a high loss.

Finally, we compare the performance of our method against four baselines in terms of test misclassification error. Refer to Appendix C for more details on the baselines, which we refer to as confidence-based triage [5], score-based triage [1], surrogate-based triage [6] and full automation triage[10] . Figure 4 summarizes the results. We find that, in each dataset, our algorithm is able to find the predictive model and the triage policy with the lowest misclassification across the entire span of $b$ values. One could view these values of maximum triage levels as the optimal automation levels (within the level values we experimented with).

## 7  Conclusions

In this paper, we have contributed towards a better understanding of supervised learning under algorithmic triage. We have first identified under which circumstances predictive models may benefit from algorithmic triage, including those trained for full automation. Then, given a predictive model and desired level of triage, we have shown that the optimal triage policy is a deterministic threshold rule in which triage decisions are derived deterministically from the model and human per-instance errors. Finally, we have introduced a practical algorithm to train supervised learning models under triage and have shown that it outperforms several competitive baselines.

Our work also opens many interesting venues for future work. For example, we have assumed that each instance is predicted by either a predictive model or a human expert. However, there may be many situations in which human experts predict all instances but their predictions are informed by a predictive model [38]. We have shown that our algorithm is guaranteed to converge to a local minimum of the empirical risk, however, it would be interesting to analyze the convergence rate and the generalization error. Finally, it would be valuable to assess the performance of supervised learning models under algorithmic triage using interventional experiments on a real-world application.

---

[10]Among all the baselines we are aware of [1–6], we did not compare with De et al. [2, 3] because they only allow linear models and SVMs and we did not compare with Wilder et al. [4] because they did not provide enough details to implement their method.

**Acknowledgment.** Okati and Gomez-Rodriguez acknowledge support from the European Research Council (ERC) under the European Union's Horizon 2020 research and innovation programme (grant agreement No. 945719). De has been partially supported by a DST Inspire Faculty Award.

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
