# A  Proofs

**Proof of Proposition 1.** Due to Jensen's inequality and the fact that, by assumption, the distribution of human predictions $P(h \mid \mathbf{x})$ is not a point-mass, it holds that $\mathbb{E}_h[\ell(h(\mathbf{x}), y) \mid \mathbf{x}] > \ell(\mu_h(\mathbf{x}), y)$. Hence,

$$\mathbb{E}_{\mathbf{x},y,h}\left[(1 - \pi(\mathbf{x}))\,\ell(m(\mathbf{x}), y) + \pi(\mathbf{x})\,\ell(h(\mathbf{x}), y)\right] > \mathbb{E}_{\mathbf{x},y}\left[(1 - \pi(\mathbf{x}))\,\ell(m(\mathbf{x}), y) + \pi(\mathbf{x})\,\ell(\mu_h(\mathbf{x}), y)\right]. \tag{12}$$

**Proof of Proposition 2.** Let $\pi(\mathbf{x}) = \mathbb{I}(\mathbf{x} \in \mathcal{V})$. Then, we have:

$$
\begin{aligned}
L(\pi, m_0^*) &= \int_{\mathbf{x} \in \mathcal{X} \setminus \mathcal{V}} \mathbb{E}_{y|\mathbf{x}}\left[\ell(m_0^*(\mathbf{x}), y)\right]\,dP + \int_{\mathbf{x} \in \mathcal{V}} \mathbb{E}_{y,h|\mathbf{x}}\left[\ell(h, y)\right]\,dP \\
&\stackrel{(i)}{<} \int_{\mathbf{x} \in \mathcal{X} \setminus \mathcal{V}} \mathbb{E}_{y|\mathbf{x}}\left[\ell(m_0^*(\mathbf{x}), y)\right]\,dP + \int_{\mathbf{x} \in \mathcal{V}} \mathbb{E}_{y|\mathbf{x}}\left[\ell(m_0^*(\mathbf{x}), y)\right]\,dP \\
&= \int_{\mathbf{x} \in \mathcal{X}} \mathbb{E}_{y|\mathbf{x}}\left[\ell(m_0^*(\mathbf{x}), y)\right]\,dP \\
&\stackrel{(ii)}{=} L(\pi_0, m_0^*),
\end{aligned}
$$

where inequality $(i)$ holds by assumption and equality $(ii)$ holds by the definition of $\pi_0(\mathbf{x})$.

**Proof of Theorem 3.** We first provide the proof of the unconstrained case. First, we note that,

$$
\begin{aligned}
L(\pi, m) &= \mathbb{E}_{\mathbf{x},h}\left[(1 - \pi(\mathbf{x}))\,\mathbb{E}_{y|\mathbf{x}}[\ell(m(\mathbf{x}), y)] + \pi(\mathbf{x})\,\mathbb{E}_{y|\mathbf{x}}[\ell(h, y)]\right] \\
&= \mathbb{E}_{\mathbf{x}}\left[(1 - \pi(\mathbf{x}))\,\mathbb{E}_{y|\mathbf{x}}[\ell(m(\mathbf{x}), y)] + \pi(\mathbf{x})\,\mathbb{E}_{y,h|\mathbf{x}}[\ell(h, y)]\right] \\
&= \mathbb{E}_{\mathbf{x}}\left[\pi(\mathbf{x})\Big[\mathbb{E}_{y,h|\mathbf{x}}[\ell(h, y)] - \mathbb{E}_{y|\mathbf{x}}[\ell(m(\mathbf{x}), y)]\Big]\right] + \mathbb{E}_{\mathbf{x},y}[\ell(m(\mathbf{x}), y)]
\end{aligned}
$$

Since the second term in the above equation does not depend on $\pi$, we can find the optimal policy $\pi$ by solving the following optimization problem:

$$
\begin{aligned}
\underset{\pi}{\text{minimize}} \quad & \mathbb{E}_{\mathbf{x}}\left[\pi(\mathbf{x})\Big[\mathbb{E}_{y,h|\mathbf{x}}[\ell(h, y)] - \mathbb{E}_{y|\mathbf{x}}[\ell(m(\mathbf{x}), y)]\Big]\right] \\
\text{subject to} \quad & 0 \le \pi(\mathbf{x}) \le 1 \quad \forall\, \mathbf{x} \in \mathcal{X}.
\end{aligned}
$$

Note that the above problem is a linear program and it decouples with respect to $\mathbf{x}$. Therefore, for each $\mathbf{x}$, the optimal solution is clearly given by:

$$\pi_m^*(d = 1 \mid \mathbf{x}) = \begin{cases} 1 & \text{if } \mathbb{E}_{y|\mathbf{x}}[\ell(m(\mathbf{x}), y) - \mathbb{E}_{h|\mathbf{x}}[\ell(h, y)]] > 0 \\ 0 & \text{otherwise} \end{cases}$$

Next, we provide the proof of the constrained case. Here, we need to solve the following optimization problem:

$$
\begin{aligned}
\underset{\pi}{\text{minimize}} \quad & \mathbb{E}_{\mathbf{x}}\left[\pi(\mathbf{x})\Big[\mathbb{E}_{y,h|\mathbf{x}}[\ell(h, y)] - \mathbb{E}_{y|\mathbf{x}}[\ell(m(\mathbf{x}), y)]\Big]\right] \\
\text{subject to} \quad & \mathbb{E}_{\mathbf{x}}[\pi(\mathbf{x})] \le b, \\
& 0 \le \pi(\mathbf{x}) \le 1 \quad \forall\, \mathbf{x} \in \mathcal{X}.
\end{aligned}
$$

To this aim, we consider the dual formulation of the optimization problem, where we only introduce a Lagrangian multiplier $\tau_{P,b}$ for the first constraint, *i.e.*,

$$
\begin{aligned}
\underset{\tau_{P,b} \ge 0}{\text{maximize}}\,\underset{\pi}{\text{minimize}} \quad & \mathbb{E}_{\mathbf{x}}\left[\pi(\mathbf{x})\Big[\mathbb{E}_{y,h|\mathbf{x}}[\ell(h, y)] - \mathbb{E}_{y|\mathbf{x}}[\ell(m(\mathbf{x}), y)]\Big]\right] \\
& + \mathbb{E}_{\mathbf{x}}\left[\tau_{P,b}\,(\pi(\mathbf{x}) - b)\right] \tag{13} \\
\text{subject to} \quad & 0 \le \pi(\mathbf{x}) \le 1 \quad \forall\, \mathbf{x} \in \mathcal{X}. \tag{14}
\end{aligned}
$$

The inner minimization problem can be solved using the similar argument for the unconstrained case. Therefore, we have:

$$\pi_{m^*,b}(d = 1 \mid \mathbf{x}) = \begin{cases} 1 & \text{if } \mathbb{E}_{y|\mathbf{x}}[\ell(m(\mathbf{x}), y) - \mathbb{E}_{h|\mathbf{x}}[\ell(h, y)]] > t_{P,b,m} \\ 0 & \text{otherwise} \end{cases}$$

where

$$t_{P,b} = \operatorname*{argmax}_{\tau_{P,b} \geq 0} \mathbb{E}_{\mathbf{x}} \left[ \min \left( \mathbb{E}_{y|\mathbf{x}}[\mathbb{E}_{h|\mathbf{x}}[\ell(h, y)] - \ell(m(\mathbf{x}), y)] + \tau_{P,b},\, 0 \right) - \tau_{P,b}\, b \right]$$

**Proof of Proposition 4.** The optimal predictive model $m_{\theta_0^*}$ under full automation within a parameterized hypothesis class of predictive models $\mathcal{M}(\Theta)$ satisfies that

$$\nabla_\theta \, L(\pi_0, m_\theta)|_{\theta=\theta_0^*} = \mathbb{E}_{\mathbf{x},y} \left[ \nabla_\theta \, \ell(m_\theta(\mathbf{x}), y)|_{\theta=\theta_0^*} \right] = \mathbf{0} \tag{15}$$

and the optimal predictive model $m_{\theta^*}$ under $\pi_{m_{\theta^*},b}^*$ satisfies that

$$\nabla_\theta \, L(\pi_{m_\theta,b}^*, m_\theta)\big|_{\theta=\theta^*} = \mathbf{0}. \tag{16}$$

Now we have that

$$\begin{aligned}
\nabla_\theta \, L(\pi_{m_\theta,b}^*, m_\theta)\big|_{\theta=\theta_0^*} &= \nabla_\theta \, L(\pi_0, m_\theta)|_{\theta=\theta_0^*} \\
&\quad - \nabla_\theta \, \mathbb{E}_{\mathbf{x}} \left[ \text{THRES}_{t_{P,b,m}} \left( \mathbb{E}_{y|\mathbf{x}} \left[ \ell(m(\mathbf{x}), y) - \mathbb{E}_{h|\mathbf{x}}[\ell(h, y)] \right], 0 \right) \right]\big|_{\theta=\theta_0^*} \\
&= 0 - \int_{\mathbf{x} \in \mathcal{V}} \mathbb{E}_{y|\mathbf{x}} \left[ \nabla_\theta \ell(m_\theta(\mathbf{x}), y)|_{\theta=\theta_0^*} \right] dP - \int_{\mathbf{x} \in \mathcal{X} \setminus \mathcal{V}} 0 \, dP \neq \mathbf{0}. \tag{17}
\end{aligned}$$

where we have used that

$$\nabla_x \text{THRES}_{t_{P,b,m}}(f(x), 0) = \begin{cases} \nabla_x f(x) & \text{if } f(x) > t_{P,b,m} \\ 0 & \text{if } f(x) < t_{P,b,m}. \end{cases}$$

Hence, we can immediately conclude that $L(\pi_{m_{\theta_0^*},b}^*, m_{\theta_0^*}) > \min_{\theta \in \Theta} L(\pi_{m_\theta,b}^*, m_\theta)$.

**Proof of Proposition 5.** Under triage policy $\pi_{m_{\theta'},b}^*$, we have that:

$$\begin{aligned}
\nabla_\theta \, L(\pi_{m_{\theta'},b}^*, m_\theta)\Big|_{\theta=\theta'} &= \nabla_\theta \, \mathbb{E}_{\mathbf{x}} \left[ (1 - \pi_{m_{\theta'}}^*(\mathbf{x})) \, \mathbb{E}_{y|\mathbf{x}}[\ell(m_\theta(\mathbf{x}), y)] + \pi_{m_{\theta'},b}^*(\mathbf{x}) \, \mathbb{E}_{y,h|\mathbf{x}}\left[\ell(h, y)\right] \right]\Big|_{\theta=\theta'} \\
&= \mathbb{E}_{\mathbf{x}} \left[ (1 - \pi_{m_{\theta'},b}^*(\mathbf{x})) \, \mathbb{E}_{y|\mathbf{x}}[\nabla_\theta \, \ell(m_\theta(\mathbf{x}), y)|_{\theta=\theta'}] \right] \\
&= \int_{\mathbf{x} \in \mathcal{V}} \mathbb{E}_{y|\mathbf{x}} \left[ \nabla_\theta \ell(m_\theta(\mathbf{x}), y)|_{\theta=\theta'} \right] \neq \mathbf{0},
\end{aligned}$$

where $\mathcal{V} = \{\mathbf{x} \mid \pi_{m_{\theta'},b}(\mathbf{x}) = 0\}$. Hence, we can immediately conclude that $L(\pi_{m_\theta b}^*, m_{\theta'}) > \min_{\theta \in \Theta} L(\pi_{m_\theta,b}^*, m_\theta)$.

**Proof of Proposition 6.** Since $\pi_{m_{\theta_t},b}^* = \operatorname{argmin}_\pi L(\pi, m_{\theta_t})$, we have that:

$$L(\pi_{m_{\theta_t},b}^*, m_{\theta_t}) \leq L(\pi_{m_{\theta_{t-1}},b}^*, m_{\theta_t}) \tag{18}$$

Then, if $\theta_t^{(i)}$ is computed from $\theta_t^{(i-1)}$ using Eq. 8, then we have that [39, Eq. 9.17]:

$$\begin{aligned}
L(\pi_{m_{\theta_{t-1}},b}^*, m_{\theta_t^{(i)}}) &\leq L(\pi_{m_{\theta_{t-1}},b}^*, m_{\theta_t^{(i-1)}}) \\
&\quad + \nabla_\theta L(\pi_{m_{\theta_{t-1}},b}^*, m_{\theta_t^{(i-1)}})^\top (\theta_t^{(i)} - \theta_t^{(i-1)}) + \frac{\Lambda}{2} \left\| \theta_t^{(i-1)} - \theta_t^{(i)} \right\|^2 \\
&\overset{(a)}{=} L(\pi_{m_{\theta_{t-1}},b}^*, m_{\theta_t^{(i-1)}}) - \alpha^{(i-1)} \nabla_\theta L(\pi_{m_{\theta_{t-1}},b}^*, m_{\theta_t^{(i-1)}})^\top \nabla_\theta L(\pi_{m_{\theta_{t-1}},b}^*, m_{\theta_t^{(i-1)}}) \\
&\quad + (\alpha^{(i-1)})^2 \frac{\Lambda}{2} \left\| \nabla_\theta L(\pi_{m_{\theta_{t-1}}}^*, m_{\theta_t^{(i-1)}}) \right\|^2 \\
&= L(\pi_{m_{\theta_{t-1}},b}^*, m_{\theta_t^{(i-1)}}) - \left( \alpha^{(i-1)} - (\alpha^{(i-1)})^2 \frac{\Lambda}{2} \right) \left\| \nabla_\theta L(\pi_{m_{\theta_{t-1}},b}^*, m_{\theta_t^{(i-1)}}) \right\|^2 \\
&\overset{(b)}{<} L(\pi_{m_{\theta_{t-1}},b}^*, m_{\theta_t^{(i-1)}}) - \frac{\alpha^{(i-1)}}{2} \left\| \nabla_\theta L(\pi_{m_{\theta_{t-1}},b}^*, m_{\theta_t^{(i-1)}}) \right\|^2 \\
&< L(\pi_{m_{\theta_{t-1}},b}^*, m_{\theta_t^{(i-1)}}), \tag{19}
\end{aligned}$$

where equality $(a)$ follows from the fact that

$$\theta_t^{(i)} - \theta_t^{(i-1)} = -\alpha^{(i-1)}\nabla_\theta \left. L(\pi^*_{m_{\theta_{t-1}},b}, m_\theta)\right|_{\theta=\theta_t^{(i-1)}} \tag{20}$$

and inequality $(b)$ follows by assumption, *i.e.*, $\alpha^{(i-1)}\Lambda < 1$.

Eq. 19 directly implies that

$$L(\pi^*_{m_{\theta_{t-1}},b}, m_{\theta_t}) < L(\pi^*_{m_{\theta_{t-1}},b}, m_{\theta_t^{(0)}}) = L(\pi^*_{m_{\theta_{t-1}},b}, m_{\theta_{t-1}}),$$

where the last equality follows by assumption, *i.e.*, $\theta_t^{(0)} = \theta_{t-1}$. This result, together with Eq. 18, proves the proposition.

**Proof of Theorem 7.** Let $\Pi_b := \{\pi \in \Pi \,|\, \mathbb{E}_\mathbf{x}\left[\pi(\mathbf{x})\right] \leq b\}$ and $\Phi_t = L(\pi^*_{m_{\theta_t},b}, m_{\theta_t}) - L(\pi^*_{m_{\theta^*},b}, m_{\theta^*})$. Then, note that, for all $\pi \in \Pi_b$, we have that

$$\nabla_\theta^2 L(\pi, m_\theta) = \mathbb{E}_{\mathbf{x},y}\left[(1-\pi(\mathbf{x}))\nabla_\theta^2 \ell(m_\theta(\mathbf{x}), y)\right] \implies \Lambda_{\min}(1-b)\mathbb{I} \preccurlyeq \nabla_\theta^2 L(\pi, m_\theta) \preccurlyeq \Lambda_{\max}\mathbb{I} \tag{21}$$

Moreover, we also have that

$$\Phi_{t+1} = L(\pi^*_{m_{\theta_{t+1}},b}, m_{\theta_{t+1}}) - L(\pi^*_{m_{\theta^*},b}, m_{\theta^*}) \overset{(i)}{\leq} L(\pi^*_{m_{\theta^*},b}, m_{\theta_{t+1}}) - L(\pi^*_{m_{\theta^*},b}, m_{\theta^*})$$

$$\overset{(ii)}{\leq} \frac{\Lambda_{\max}}{2}\|\theta_{t+1}-\theta^*\|^2, \tag{22}$$

where (i) follows from the fact that $\operatorname{argmin}_{\pi\in\Pi_b} L(\pi, m_{\theta_{t+1}}) = \pi^*_{m_{\theta_{t+1}},b}$ and (ii) follows from the Taylor series expansion of $L(\pi^*_{m_{\theta^*},b}, m_\theta)$ around $\theta = \theta^*$ and Eq. 21, *i.e.*,

$$L(\pi^*_{m_{\theta^*},b}, m_{\theta_{t+1}}) \leq L(\pi^*_{m_{\theta^*},b}, m_{\theta^*}) + \underbrace{\nabla_\theta L(\pi^*_{m_{\theta^*},b}, m_\theta)^\top|_{\theta=\theta^*}}_{=0}(\theta_{t+1}-\theta^*) + \frac{\Lambda_{\max}}{2}\|\theta_{t+1}-\theta^*\|^2.$$

Next, we have that

$$L(\pi^*_{m_{\theta_t},b}, m_{\theta_t}) \overset{(i)}{\geq} L(\pi^*_{m_{\theta_t},b}, m_{\theta_{t+1}}) + \frac{(1-b)\Lambda_{\min}}{2}\|\theta_{t+1}-\theta_t\|^2$$

$$\overset{(ii)}{\geq} L(\pi^*_{m_{\theta_{t+1}},b}, m_{\theta_{t+1}}) + \frac{(1-b)\Lambda_{\min}}{2}\|\theta_{t+1}-\theta_t\|^2, \tag{23}$$

where (i) follows from the fact that $\theta_{t+1} = \operatorname{argmin}_\theta L(\pi^*_{m_{\theta_t}}, m_\theta)$, the Taylor series expansion of $L(\pi^*_{m_{\theta_t},b}, m_\theta)$ around $\theta = \theta_{t+1}$, and Eq. 21, *i.e.*,

$$L(\pi^*_{m_{\theta_t},b}, m_{\theta_t}) \geq L(\pi^*_{m_{\theta_t},b}, m_{\theta_{t+1}}) + \underbrace{\nabla_\theta L(\pi^*_{m_{\theta_t},b}, m_\theta)^\top|_{\theta=\theta_{t+1}}}_{=0}(\theta_t-\theta_{t+1})$$

$$+ \frac{\Lambda_{\min}(1-b)}{2}\|\theta_t-\theta_{t+1}\|^2,$$

and (ii) follows from the fact that $\operatorname{argmin}_{\pi\in\Pi_b} L(\pi, m_{\theta_{t+1}}) = \pi^*_{m_{\theta_{t+1}},b}$. Then, from Eq. 23, it readily follows that

$$\Phi_t - \Phi_{t+1} \geq \frac{(1-b)\Lambda_{\min}}{2}\|\theta_{t+1}-\theta_t\|^2. \tag{24}$$

Now, combining Eq. 22 and Eq. 24, we have that

$$\Phi_{t+1} \leq \frac{\Lambda_{\max}}{2}\|\theta_{t+1}-\theta^*\|^2 \leq \Lambda_{\max}\left[\|\theta_{t+1}-\theta_t\|^2 + \|\theta_t-\theta^*\|^2\right]$$

$$\overset{(i)}{\leq} \frac{2\Lambda_{\max}}{(1-b)\Lambda_{\min}}(\Phi_t - \Phi_{t+1}) + \frac{4H^2\Lambda_{\max}}{\Lambda_{\min}^2(1-b)^2}. \tag{25}$$

where we have used Proposition 8 in (i). Finally, from Eq. 25, it readily follows that

$$\lim_{t\to\infty}\Phi_{t+1} \leq \frac{4H^2\Lambda_{\max}}{\Lambda_{\min}^2(1-b)^2}.$$

This concludes the proof.

**Proposition 8** *Let $\ell(\cdot)$ be convex with respect to $\theta$ and thus the output of the SGD algorithm $\theta_t = \arg\min_\theta L(\pi^*_{m_{\theta_{t-1}}}, m_\theta)$. Moreover, assume that $\nabla^2_\theta \ell(m_\theta(\mathbf{x}), y) \succcurlyeq \Lambda_{\min}$, with $\Lambda_{\min} > 0$, and $\ell(\overset{.}{)}$ be $H$-Lipschitz, i.e., $\ell(m_\theta(\mathbf{x}), y) - \ell(m_{\theta'}(\mathbf{x}), y) \leq H \cdot \|\theta - \theta'\|$. Then, we have $\|\theta_t - \theta^*\| \leq \frac{2H}{\Lambda_{\min}(1-b)}$.*

**Proof** We have that

$$\frac{\Lambda_{\min}(1-b)}{2} \|\theta_t - \theta^*\|^2 \overset{(i)}{\leq} L(\pi^*_{m_{\theta^*},b}, m_{\theta_t}) - L(\pi^*_{m_{\theta^*},b}, m_{\theta^*}) \leq H \|\theta_t - \theta^*\|, \tag{26}$$

where (i) follows from the Taylor series expansion of $L(\pi^*_{m_{\theta^*},b}, m_\theta)$ around $\theta = \theta^*$. Then, from Eq. 26, it readily follows

$$\|\theta_t - \theta^*\| \leq \frac{2H}{\Lambda_{\min}(1-b)} \tag{27}$$

$\blacksquare$

## B    Scalability Analysis of Algorithm 1

In comparison with vanilla SGD, our algorithm just needs to additionally call the function TRIAGE before each iteration. This function first sorts the samples in the corresponding minibatch in decreasing order of the model loss minus the human loss and then returns the first $\max(\lceil(1-b)|\mathcal{D}|\rceil, p)$ samples. Overall, this adds $O(T|\mathcal{D}| \log B)$ to the overall complexity of the training procedure with respect to vanilla SGD, where $B$ is the size of the minibatch used during training, $\mathcal{D}$ is the training dataset, and $T$ is the number of steps. Furthermore, note that the function APPROXIMATETRIAGEPOLICY is called only once and use SGD to train the approximate triage policy of the last predictive model. Therefore, it does not increase the computational complexity of the overall algorithm.

## C    Additional Details About the Experiments on Real Data

In what follows, we provide additional details regarding the implementation of our method as well as the baselines for the experiments on real data:

— Our method: During training, it runs Algorithm 1. During test, it lets the humans predict any sample for which $\hat{\pi}_\gamma(\mathbf{x}) \geq \hat{p}_b$, where the threshold $\hat{p}_b$ is found using cross validation.

— Confidence-based triage [5]: During training, it first estimates the probability $P(h = y)$ that humans predict the true label. Then, it proceeds sequentially and, at each step $t$, it uses SGD to train a predictive model $m_{\theta_t}$. However, in each iteration of SGD, it only uses the $\min(\lfloor b|\mathcal{D}|\rfloor, n_c)$ training samples with the lowest value of $P(h = y) - \max_{y' \in \mathcal{Y}} P(m_\theta(\mathbf{x}) = y')$ in the corresponding mini batch, where $n_c$ is the number of training samples in the mini batch for which $P(h = y) > \max_{y' \in \mathcal{Y}} P(m_\theta(\mathbf{x}) = y')$. During test, it first sorts all the samples in increasing order of $\max_{y' \in \mathcal{Y}} P(m_\theta(\mathbf{x}) = y')$ and then lets the humans predict the first $\min(\lfloor b|\mathcal{D}|\rfloor, n_c)$ samples[11], where $n_c$ is the number of test samples for which $P(h = y) > \max_{y' \in \mathcal{Y}} P(m_\theta(\mathbf{x}) = y')$.

— Score-based triage [1]: During training, it uses SGD to train a predictive model $m_\theta$ using all the training samples. During test, it first sorts all the samples in increasing order of $\max_{y' \in \mathcal{Y}} P(m_\theta(\mathbf{x}) = y')$ and then lets the humans predict the first $\lfloor b|\mathcal{D}|\rfloor$ samples. Here, note that the method always lets the humans predict $\lfloor b|\mathcal{D}|\rfloor$ samples because its triage policy does not depend on the human loss.

— Surrogate-based triage [6]: During training, it trains a predictive model $m_\theta$, where $\pi(\mathbf{x}) = 1$ is just an extra label value $y_{\text{defer}}$, by minimizing a surrogate of the true loss function defined in Eq. 2. During test, it first sorts all the samples in increasing order of $\max_{y' \in \mathcal{Y}} P(m_\theta(\mathbf{x}) = y') - P(m_\theta(\mathbf{x}) = y_{\text{defer}})$ and then lets the human predict the first $min(\lfloor b|\mathcal{D}|\rfloor, n_c)$ samples where $n_c$ is the number of test samples for which $P(m_\theta(\mathbf{x}) = y_{\text{defer}}) > \max_{y' \in \mathcal{Y}} P(m_\theta(\mathbf{x}) = y')$.

---

[11]Here, note that the method assumes that the humans are uniformly accurate across samples, i.e., $P(h = y \,|\, \mathbf{x}) = P(h = y)$, both during training and test.

— Full automation triage: During training, it uses SGD to both train a predictive model $m_\theta$ using all training samples and an approximate triage policy $\hat{\pi}_\gamma$ that approximates the optimal triage policy $\pi^*_{m_\theta,b}$. During test, it lets the humans predict any sample for which $\hat{\pi}_\gamma(\mathbf{x}) \geq \hat{p}_b$, where the threshold $\hat{p}_b$ is found using cross validation.

In our experiments, our method and all the baselines use the hypothesis class of parameterized predictive models $\mathcal{M}(\Theta)$ parameterized by softmax distributions, *i.e.*,

$$m_\theta(\mathbf{x}) \sim p_{\theta;\mathbf{x}} = \text{Multinomial}\left([\exp(\phi_{y,\theta}(\mathbf{x}))]_{y \in \mathcal{Y}}\right),$$

where, for the nonlinearities $\phi_{\bullet,\theta}$, we use the following network architectures:

— Hatespeech dataset: we use the convolutional neural network (CNN) developed by Kim [36] for text classification, which consists of 3 convolutional layers with filter sizes $\{3, 4, 5\}$, respectively and with 300 neurons per layer. Moreover, each layer is followed by a ReLU non-linearity and a max pooling layer.

— Galaxy Zoo: we use the deep residual network developed by He et al. [37]. To this end, we first downsample each RGB channel of each of the images to size $224 \times 224$ and standardize its values[12]. The wide residual network consists of 50 convolutional layers. The first layer is a $7 \times 7$ convolutional layer followed by a $3 \times 3$ max pooling layer. The next 48 convolutional layers have filter sizes of $1 \times 1$ or $3 \times 3$ which are followed by an average pooling layer. The last layer is a fully connected layer. Each convolution layer is followed by ReLU nonlinearity.

In our method and all the baselines except surrogate-based triage, we use the cross-entropy loss and implement SGD using Adam optimizer [40] with initial learning rate set by cross validation independently for each method and level of triage $b$. In surrogate-based triage, we use the loss and optimization method used by the authors in their public implementation. Moreover, we use early stopping with the patience parameter $e_p = 10$, *i.e.*, we stop the training process if no reduction of cross entropy loss is observed on the validation set. Finally, to avoid that the cross entropy loss $\ell(\hat{y}, y)$ becomes unbounded whenever an instance is assigned to a human expert and all human experts predicted the same label for that instance in our dataset, we do add/substract an $\epsilon$ value to the estimated values of the conditional probabilities $P(h \mid \mathbf{x})$.

---

[12]`https://pytorch.org/hub/pytorch_vision_resnet/`

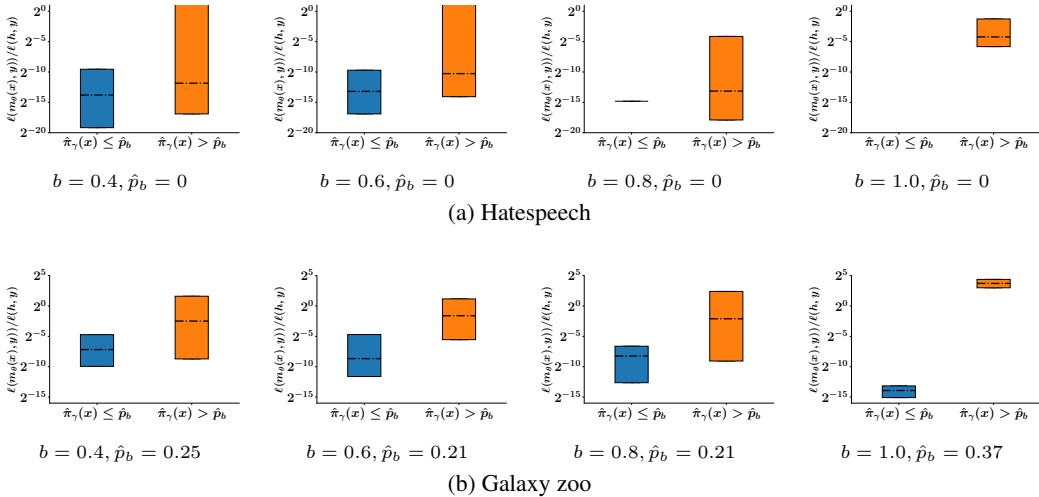

Figure 5: Ratio of model and human losses for test samples predicted by the model and test samples predicted by the humans, as dictated by the approximate triage policy $\hat{\pi}_\gamma$, for different values of the maximum level of triage $b$. In each panel, the threshold $\hat{p}_b$ is found using cross validation. Boxes indicate 25% and 75% quantiles and the horizontal lines indicate median values.

## D    Additional Evaluation of the Approximate Triage Policy

Figure 5 shows the ratio of model and human losses for those test samples predicted by the model and test samples predicted by the humans, as dictated by the approximate triage policy $\hat{\pi}_\gamma$, for different values of the maximum level of triage $b$. We find several interesting insights. We observe that the approximate triage policy $\hat{\pi}_\gamma$ lets the humans predict those samples whose ratio of model and human losses is higher, as one could have expected. Moreover, in the Hatespeech dataset, we find that the triage policy lets humans predict (almost) all the samples whenever $b = 1$ ($b = 0.8$), *i.e.*, the budget constraint in the optimization problem defined by Eq. 1 is active. This suggests that the humans are more accurate than the predictive model throughout the entire feature space. In contrast, in the Galaxy zoo dataset, the triage policy does not rely on the human predictions for all samples for $b = 1$. This suggests that the humans are less accurate than the predictive model in some regions of the feature space.