# OpenReview forum: "Differentiable Learning Under Triage"
_NeurIPS.cc/2021/Conference — NeurIPS 2021 Poster_

### Official Review · Reviewer_nQQC · 2021-07-07

**Rating:** 6
**Confidence:** 3

**Summary:**

This paper presents a method triaging  / deferring examples to human labellers in machine learning systems. Their approach is conceptually simple: begin with a model trained on the full dataset and then iteratively improve the triaged model by updating it on a subset of the data that excludes examples where the previous iteration was worst compared to the human labellers. The approach is justified through a series of theorems showing that, as long as there exist parts of the input space for which the human labelers outperform the model, then triaging offers the possibility of improving the (combined model & human) performance; and further, that gradient descent can improve the model on the restricted space implied by the thresholding approach.

This is not my area, but I generally found the paper to be clear and well written (aside from style issues discussed below).

**Limitations And Societal Impact:**

These are relatively small issues, but I would have liked to see some discussion on them:

[Interpolation regimes] Modern deep networks are often massively over-parameterized such that they can interpolate the training data (driving the loss essentially to zero on each training example); essentially memorizing training examples, while still generalizing. It seems like the method would fail in this setting because you wouldn't threshold anything? Can the approach be generalized to select which examples to triage on a validation set to avoid this issue?

[Non-convexity] It is not clear how the sequential approach that is presented here interacts with non-convexity? Is there a concern that starting the optimal model on the full dataset and then thresholding can lead you to a worse solution than you would have found if you could optimize for the thresholded loss directly? I see this isn't an issue in the convex setting, but I'm curious whether you've thought about it in the non-convex setting?


**Main Review:**

**[Building a sequence of triage policies]** I understand that we can sequentially improve the combined policy of the models an humans up to our budget by retraining after excluding the subset of examples that the previous iteration performed worst on, but, assuming I know $b$, it wasn't clear whether that sequence is of length two (i.e. train and unconstrained model and then retrain with the subset implied by $\max(\lceil(1−b) |D|\rceil,p)$), or whether I sequentially retrain with a sequence of $b_i \leq b$? And if so, how do I choose that sequence?

**[Evaluation]** I really would have liked to see more than two datasets evaluated. This is a generic method that seems to apply to any dataset, so aside from compute requirements, there doesn't seem to be anything preventing an evaluation that check whether the observed improvements over prior approaches hold across a statistically significant number of datasets?

**["Mathiness" style issues]** While the prose is generally well-written, the presentation of the propositions in section 3 (and to a lesser extent in the other sections), runs dangerously close to what Lipton & Steinhardt [2018] call 'mathiness' to "bulldoze rather than to clarify". I don't think that the authors intended this---and these comments haven't influenced my review score---but the paper could be a lot clearer if the propositions included a sentence or two on why each statement is true. For example, Prop 4 is prefaced with, "we can *identify the circumstances* under which the optimal predictive model... is suboptimal under algorithmic triage. Formally, *our main result* is the following Proposition..." (my emphasis). That sounds mysterious - when the "circumstances" are simple - as long as the thresholded gradients don't sum to zero, the model is suboptimal under triage. Making it clear that you're mostly just using the fact that loss and gradients on a subspace are typically not the same as those on the whole space would make the whole presentation easier to follow.

To be clear - I'm very happy for theorems to formally delineate an argument, but as far as possible, we should use them to tell the reader why something is true, not just what is true.

**Time Spent Reviewing:**

4

---

> ### Author Response · Authors · 2021-08-05
> **More discussion on how the sequence of machine models and triage policies are built, comments on evaluation setup and non-convexity issues, and incorporating suggestions**
>
> We thank the reviewer for the helpful comments and suggestions, which will help improve our paper.
>
> [Building a sequence of triage policies] We sequentially train the predictive models as follows. At step $t = 1$, we train the model $m_{\theta_1}$ using all training samples, as implied by the choice of initial triage policy $\pi_{0}$ in line 140. After that, at each step $t > 1$, we train the predictive model $\theta _t$ by re-training the predictive model $\theta _{t-1}$, as noted in line 147, using the $max(\lceil(1−b)|D|\rceil,p))$ training samples with the lowest value of the model loss $\ell(m _{\theta _{t-1}}(x), y)$ minus the human loss $l(h, y)$, where note that: (i) we use the predictive model $m _{\theta _{t-1}}$ fitted in step $t-1$ to calculate the model loss $\ell(m _{\theta _{t-1}}(x), y)$, as noted in line 156, and (ii) the number of training samples $max(\lceil(1−b)|D|\rceil,p))$ at each step depends on the difference of losses through $p$, which is the number of samples with negative difference of losses. We will clarify this in the revised version of the paper.
>
> In addition, note that, as argued in lines 153-159, the above procedure is equivalent to finding the predictive model that minimizes the objective function $L(\pi^* _{\theta _{t-1}}, m _{\theta _t})$ with respect to the predictive model $m _{\theta _t}$ under the constraint $E[\pi^* _{\theta _{t-1}}(x)] <= b$. Put differently, it is equivalent to solving the constrained problem defined by Eq. 1 after fixing the triage policy to the optimal triage policy under the model $m _{\theta _{t-1}}$ fitted in step $t-1$. Finally, note that Proposition 6 shows that our algorithm finds triage policies and predictive models that are guaranteed to improve in each step.
>
>
> [Evaluation] As explicitly noted in footnote 7 on page 7, we only experimented with two datasets because they were the only publicly available datasets that we found containing multiple human predictions per instance, necessary to estimate the human loss at an instance level, and a relatively large number of instances. In this context, please, note that other publicly available datasets such as Messidor or Stare used by previous work [2, 3] contain only one human prediction per instance. As a result, previous work simulated the value of human loss at an instance level.
>
>
> [Mathiness] To increase readability, for each proposition in Section 3, we will include a sentence or two on why each statement is true in the revised version of the paper, following the reviewer's suggestion.
>
>
> [Interpolation regimes] At each step t, the implementation of our algorithm that we used in the experiments only does one pass through the training data (please refer to lines 13-22 in Algorithm 1 in Appendix B). As a result, at early steps $t$, it is unlikely that the training error of the predictive model $\theta _t$ is zero on the training samples that the triage policy lets the model predict. However, at later steps $t$, as the predictive model $\theta _t$ learns to specialize in a region of the feature space---the region picked by the triage policy $\pi^* _{\theta _{t-1}}$---its training error on the samples in this region converges to zero, as shown in Figure 3. We will discuss this in the revised version of the paper.
>
>
> [Non-convexity] Please note that Proposition 6 does not require the loss to be convex and it shows that our algorithm finds triage policies and predictive models that are guaranteed to improve at each step $t$. Also please note that at each step we do optimize for the thresholded loss defined in Eq. 5 directly.

---

### Official Review · Reviewer_3WRj · 2021-07-10

**Rating:** 6
**Confidence:** 3

**Summary:**

The paper proposes a methodology for learning models when some of the prediction tasks are deferred to human experts (i.e., triage).
In the paper, the authors: (i) first derive the optimal level of triage for a given a model; (ii) then, they provide conditions under which models learnt under full automation will achieve lower predictive performance in presence of triage; (iii) then, they propose an iterative procedure to learn model and optimal level of triage jointly; (iv) lastly, they empirically verify the proposed algorith via a simulation and two case studies of real-world datasets.
The key contributions of this paper are: (1) theorem 3, i.e., the analytical derivation of the optimal triage level; (2) the proposed methodology to learn the model under triage.

**Limitations And Societal Impact:**

The authors do not discuss the potential negative societal impact of their work.

**Main Review:**

Overall, I found the paper to be well written, except for some typos and lack of clarity in the figures (see below). In particular, I found the introduction of the framework to be very clear (initial part of section 2).
The results were interesting--especially theorem 3--and the derivation of most of them was fairly simple, which is a plus.
The results also seem to be novel, but I am not fully aware of all works in this part of the literature and in particular I only have passing knowledge of [6].

One minor concern: I am confused by the definition of \mu_h(x) in line 90-91: One may be interested in the majority prediction (in case of binary classification) or in the average prediction (in case of regression), so I do not fully understand why \mu_h is defined as the argmin of the same loss (with respect to h) that appears in (2). In addition, the proof of proposition 1 seems to rely exactly on \mu_h(x)=E[h(x)], which I think it may not correspond to the \mu_h as defined in lines 90-91 (e.g., it does for the squared loss but may not for other types). That said, I found the result in this proposition to be interesting.

Other minor issues and suggestions:
* line 28: reference [7] is not about models having higher predictive performance than humans
* lines 34-36: the sentence seems slightly confusing
* line 39: algorithm*s*
* when the paper is the subject of the sentence, sometimes it is taken as singular and other times as a plural noun (line 40 vs. 42)
* another perhaps useful reference for the learning to defer literature is [Madras et at., 2017]
* line 107: it become*s*
* line 147: the loss should decrease, not increase (the inequality in the proof seems to be in the correct direction)
* figure 1: the caption is fairly clear (if one also carefully reads the main text), but the legend is not. For example, do the four columns correspond to models 1-4? In particular, I found confusing the fact that in this figure the color refers the noise whereas in figure 3 it refers to the triage level. There is also a similar issue for the tone
* line 210: there is no panel (a) in figure 1

Madras, David, Toniann Pitassi, and Richard Zemel. "Predict responsibly: improving fairness and accuracy by learning to defer." arXiv preprint arXiv:1711.06664 (2017).

**Time Spent Reviewing:**

4

---

> ### Author Response · Authors · 2021-08-05
> **Fixing inconsistency in Proposition 1, correcting minor issues and incorporating suggestions**
>
> We thank the reviewer for spotting the inconsistency in the definition of $\mu _h$ in lines 90--91, which considers the point estimate $\mu _h(x)$ with respect to the loss $\ell(h, \mu _h)$, and the proof of proposition 1, which considers the point estimate $\mu _h(x) = E[h | x]$ with respect to the quadratic loss $(h - \mu _h)^2$.
>
> In the revised version of the paper, we will change $\ell$ in line 91 to $\ell’$ and specify that $\ell’$ is a general loss function, e.g., quadratic loss in a regression task, as suggested by the reviewer. Subsequently, we will replace the statement in line 91--92, i.e.,  “However, the resulting objective would have a bias term, as formalized by the following proposition” with “However, the resulting objective would have a bias term, as formalized by the following proposition for the quadratic loss $\ell'(h, \mu _h) = (h - \mu _h)^2$. Consequently, in the statement of Proposition 1, we will also specify that it refers to the quadratic loss $\ell'(h, \mu _h) = (h - \mu _h)^2$.
>
> We thank the reviewer for all the minor issues and suggestions. In the revised version of the paper, we will address all the minor issues and incorporate all the suggestions, including the citation to [Madras et al., 2017]. In that context, we would also like to clarify that ​​the four columns indeed correspond to models 1-4. We will explicitly note that in the revised paper.
>
> We could not think of any potential negative societal impact of the work. If the reviewer(s) (or the AC) finds any potential negative social impact, we would be happy to include it in the revised version of the paper.

---

### Official Review · Reviewer_Z9up · 2021-07-16

**Rating:** 7
**Confidence:** 3

**Summary:**

This paper considers the setting of "algorithmic triage", where a predictive model can choose to route some data points to a human expert for manual review.  The authors formally define the settings under which triage is useful, and show a motivating result that training a model for non-triage settings may be sub-optimal for learning under triage.  They then derive the optimal triage policy as a deterministic threshold rule, show a gradient-based algorithm for this, and demonstrate strong empirical performance on several synthetic and real datasets.

**Main Review:**

This paper proposes an approach to algorithmic triage where the goal is to learn both (i) a predictive model (ie the standard classifier) and *also* (ii) a triage policy model which determines whether the predictive model or a human expert should label a given instance, by optimizing the overall loss subject to some limit on human annotator time.  The paper:
- (i) Formalizes the notion that expert annotator uncertainty, as a function of the data, needs to be taken into account to optimize the loss in Prop. 1
- (ii) Establishes formally that a model worse than human experts on some instances will always benefit from an optimal triage policy (which seems like a trivial statement...?)
- (iii) Establishes that the optimal model without triage is *not* the optimal model under triage- a very interesting result and great motivation for this line of work!
- (iv) Proposes an iterative gradient/MC-based algorithm for estimating the model and triage policy pair, which they show in Prop. 6 is locally optimal under mild conditions
- (v) Validates the approach on a range of synthetic and empirical experiments

Overall this is a strong and thoroughly written paper that motivates the setting well, proposes a sensible algorithmic approach, and shows strong empirical results.  Weaknesses include the fact that there is only local convergence shown formally for the actual proposed algorithm; the experimental setup could be better described, especially the description of exact implementation of baselines; and more ablations exposing limitations of the proposed method would have been very helpful.

**Time Spent Reviewing:**

3

---

> ### Author Response · Authors · 2021-08-05
> **Global guarantees, description of the experimental setup and ablation study**
>
> We thank the reviewer for the helpful comments and suggestions, which will help improve our paper.
>
> Since we submitted our paper, we have managed to prove that, under some relatively standard technical conditions, our algorithm enjoys nontrivial global guarantees whenever the loss $\ell$ is convex. More specifically, we have the following theorem:
>
> Theorem: Let $\ell(\cdot)$ be convex in $\theta$ and thus the output  of the SGD algorithm
> $\theta _t =  \text{argmin} _{\theta}  L(\pi^* _{m _{\theta _{t-1}},b}, m _{\theta}) $.
> Moreover, assume that $\Lambda _{\text{min}}\mathbb{I} \preccurlyeq \nabla^2 \ell(m _{\theta}(\mathbf{x}),y) \preccurlyeq \Lambda _{\text{max}} \mathbb{I}$,  with $\Lambda _{\text{min}} > 0$, and $\ell ( \cdot)$ be $H$-Lipschitz, i.e., $\ell(m _{\theta}(\mathbf{x}),y)-\ell(m
>  _{\theta'}(\mathbf{x}),y) \le H\cdot \big|\big|\theta-\theta'\big|\big|$. Then, we have that
>
>  $\lim _{t \to \infty} L( \pi^* _{m _{\theta _{t}},b},m _{\theta _{t}})-L( \pi^* _{m _{\theta^*},b}, m _{\theta^*}) \le  \frac{4H^2 \Lambda _{\text{max}}}{\Lambda _{\text{min}}^{2} (1-b)^{2}}.$
>
> Proof: Let $\Pi_b :=  \left( \pi \in \Pi \| \mathbb{E} _{\mathbf{x}}\left[\pi(\\mathbf{x})\right] \le b \right) $ and $\Phi _t =L(\pi^* _{m _{\theta _{t}}, b}, m _{\theta _{t}})-L(\pi^* _{m  _{\theta^*}, b}, m _{\theta^*})$. Then, note that, for all $\pi \in \Pi_b$, we have that
> $\nabla _{\theta} ^2  L(\pi,m _{\theta}) = \mathbb{E} _{\mathbf{x}, y} \left[ (1-\pi(\mathbf{x}))  \nabla^2 \ell(m _{\theta}(\mathbf{x}), y)  \right]
> \implies \Lambda _{\text{min}}(1-b) \mathbb{I} \preccurlyeq \nabla _{\theta} ^2 L(\pi,m _{\theta}) \preccurlyeq  \Lambda _{\text{max}} \mathbb{I}\qquad (1) $.
>
> Moreover, we also have that:
> $ \Phi _{t+1} = L( \pi^*  _ {m _{\theta _{t+1}},b},m _{\theta _{t+1}})-L( \pi^* _{m _{\theta^*},b},m _{\theta^*})
> \overset{(i)}{\le}  L( \pi^*  _ {m _{\theta^*},b} ,m _{\theta _{t+1}})-L( \pi^* _{m _{\theta^*},b},m _{\theta^*})
> \overset{(ii)}{\le} \frac{\Lambda _{max}}{2} ||{\theta _{t+1}-\theta^*}||^2 \qquad (2)$,
>
> where (i) follows from the fact that $\text{argmin} _{\pi\in\Pi _b} L(\pi,m _{\theta _{t+1}}) = \pi^* _{m _{\theta _{t+1}},b}$
> and (ii) follows from the Taylor series expansion of $L(\pi^* _{m _{\theta^*},b} ,m _{\theta})$ around $\theta = \theta^*$ and Eq. 1,  i.e.,
> $ L( \pi^* _{m _{\theta^*},b} , m _{\theta _{t+1}}) \le L( \pi^* _{m _{\theta^*},b} ,m _{\theta^*}) +
>     \underbrace{\nabla _{\theta} L( \pi^* _{m _{\theta^*},b} ,m _{\theta}) ^\top \big| _{\theta=\theta^*} } _{=0} (\theta _{t+1}-\theta^*)   +  \frac{\Lambda _{max}}{2} ||{\theta  _{t+1}-\theta^*}|| ^2 $.
>
> Similarly, we have that
> $ L( \pi^* _{m _{\theta _{t}},b},m _{\theta _{t}}) {\ge} L( \pi^* _{m _{\theta _{t}},b},m _{\theta  _{t+1}}) + \frac{(1-b)\Lambda _{min}}{2} ||{\theta _{t+1}-\theta _t}||^2
> {\ge} L( \pi^* _{m _{\theta _{t+1}},b},m _{\theta _{t+1}}) + \frac{(1-b)\Lambda _{min}}{2} ||{\theta _{t+1}-\theta _t}||^2, \qquad (3) $
>
>
> Then, from Eq. 2, it readily follows that
> $ \Phi _t-\Phi _{t+1} \ge \frac{(1-b)\Lambda _{min}}{2} ||{\theta _{t+1}-\theta _t}|| ^2.  (4)$
>
> Now, combining Eq. 2 and Eq.4, we have that
> $ \Phi _{t+1}   \leq \frac{\Lambda _{max}}{2} ||{\theta _{t+1}-\theta^*}||^2 \le \Lambda _{max} \left[ ||{\theta  _{t+1}-\theta _t}||^2 +||{\theta
>  _{t}-\theta^*}^2 || \right] $,
> which implies  $ \Phi _{t+1}  \le  \frac{2 \Lambda  _{max}}{(1-b)\Lambda  _{min} } (\Phi _t-\Phi _{t+1}) + \frac{4H^2 \Lambda _{max}}{\Lambda _{min} ^{2} (1-b)^{2}} \quad (5)$.
>
> The last inequality is due to:
> $ \frac{\Lambda _{min} (1-b)}{2} ||{\theta _{t}-\theta^*}||^2
> \overset{(i)}{\le} L( \pi^* _{m _{\theta^*},b} ,m _{\theta _{t}})-L( \pi^* _{m _{\theta^*},b},m _{\theta^*}) \le H ||{\theta _{t}-\theta^*}|| $. Now by putting $t\to \infty$ into Eq. 5, we have the required bound.
>
> We would be happy to include such an additional theoretical result in the revised paper if the reviewers and AC deem this is appropriate at this point.
>
>
> Due to space constraints, we have deferred part of the description of the experimental setup to Appendix C in the supplementary material. This includes an in-depth description of the exact implementation of the baselines we compared with. To further facilitate reproducibility, we also plan to release a public implementation of our method, the exact implementation of the baselines and the post processed datasets. To this end, we will build up on the code we included as supplementary material with our submission.
>
>
> In the revised version of our paper, we will include additional ablation experiments ​​besides the current synthetic experiments which show the performance of our algorithm in cases where the optimal triage policy is not used or where the machine model is not trained using the optimal policy.

---

### Official Review · Reviewer_E3Mu · 2021-07-20

**Rating:** 7
**Confidence:** 2

**Summary:**

This paper studies the problem of algorithmic triage, where a predictive model and human experts together generate predictions. The authors introduced a theoretical analysis to model the interplay between the prediction accuracy of the model and the human experts. The authors later introduced a gradient based algorithm to find a sequence of predictive models and triage politics of increasing performance.  Extensive experiments on both synthetic dataset and two real world datasets showed improvement of proposed method over baseline methods.

**Limitations And Societal Impact:**

Yes.

**Main Review:**

This paper studies a very interesting problem, where human experts can work together with predictive models to generate predictions. Relevant but slightly different from active learning, which tries to generate better predictive models with humans in the loop, the algorithmic triage problem targets at minimizing the overall loss with constraint to cost on human experts.

However, it might still be good to include more discussions on how active learning based methods would perform in this setup.

The paper is well written and easy to follow. The authors also provided theoretical analysis and support for their proposed method, which seems to be technically sound and efficient in providing policy and predictive models given cost constraints.

The author conducted experiments on synthetic datasets to show how the proposed method works and then showed the improvement on two real world datasets.


**Time Spent Reviewing:**

3

---

> ### Author Response · Authors · 2021-08-05
> **More discussions on active learning**
>
> We thank the reviewer for the helpful comments and suggestions, which will help improve our paper.
>
> Active learning based methods aim to find which subset of samples human experts should label so that a model trained on these subset of samples predicts accurately any sample at test time. Therefore, the resulting predictive model would perform, at best, as well as the best predictive model trained under full automation using all training samples. In contrast, in learning under triage, all training samples are labeled and the goal is to use all these samples, together with imperfect human predictions, to find the predictive model and the triage policy that would perform best under a specific automation level at test time. We will expand our discussion on how active learning based methods would perform in our problem setting in lines 70-72 of the revised paper.

---

### Decision · Program_Chairs · 2021-09-27

**Decision:**

Accept (Poster)

**Comment:**

Strengths:
- Sensible algorithmic approach
- Sound theoretical analysis for setup and algorithm
- Thorough experiments on both synthetic and real data

Weaknesses:
- Clarity could be improved, over-usage of math and notation
- In experiments - better describing baselines and adding ablation studies


Summary:

Reviewers are in agreement that this is a good paper and should be accepted. Most concerns regarded particular inclarities, but discussions were helpful in this respect. The authors also extended their local convergence result to a global results under some assumptions, which addressed a concern raised by one of the reviewers.